# Exogenous GA_3_ Enhances Nitrogen Uptake and Metabolism under Low Nitrate Conditions in ‘Duli’ (*Pyrus betulifolia* Bunge) Seedlings

**DOI:** 10.3390/ijms25147967

**Published:** 2024-07-21

**Authors:** Weilong Zhang, Xiaohua Cheng, Zhaotian Jing, Ying Cao, Shuai Yuan, Haixia Zhang, Yuxing Zhang

**Affiliations:** College of Horticulture, Hebei Agricultural University, Baoding 071001, China; 18322711826@163.com (W.Z.); chengxiaohua2016@126.com (X.C.); 13409298238@163.com (S.Y.); zhx2323a@163.com (H.Z.)

**Keywords:** plant endogenous hormone, growth regulation, N metabolism, NO_3_^−^ deficiency, pear rootstocks

## Abstract

‘Duli’ (*Pyrus betulifolia* Bunge) is one of the main rootstocks of pear trees in China. Gibberellin (GA) is a key plant hormone and the roles of GA in nitrate (NO_3_^−^) uptake and metabolism in plants remain unclear. In this study, we investigated the effects of exogenous GA_3_ on the N metabolism of ‘Duli’ seedlings under NO_3_^−^ deficiency. The results showed that exogenous GA_3_ significantly improves ‘Duli’ growth under NO_3_^−^ deficiency. On the one hand, GA_3_ altered the root architecture, increased the content of endogenous hormones (GA_3_, IAA, and ZR), and enhanced photosynthesis; on the other hand, it enhanced the activities of N−metabolizing enzymes and the accumulation of N, and increased the expression levels of N absorption (*PbNRT2*) and the metabolism genes (*PbNR*, *PbGILE*, *PbGS*, and *PbGOGAT*). However, GA_3_ did not delay the degradation of chlorophyll. Paclobutrazol had the opposite effect on growth. Overall, GA_3_ can increase NO_3_^−^ uptake and metabolism and relieve the growth inhibition of ‘Duli’ seedlings under NO_3_^−^ deficiency.

## 1. Introduction

Nitrogen (N) is not only a crucial mineral element that plays a key role in plant growth, but is also an important component of chlorophyll, amino acids, nucleic acids, and secondary metabolites [1]. In nature, most plants cannot produce N by themselves and must obtain it from the soil. N is mainly absorbed and utilized by plants in the form of nitrate (NO_3_^−^) and ammonium (NH_4_^+^), which are most common in aerobic and waterlogged soil, respectively [2]. In plants, N is absorbed by NO_3_^−^/NH_4_^+^ transporter family proteins (NRTs and AMTs) in the roots [3]. It is converted into nitrite (NO_2_^−^) in the cytosol by NO_3_^−^ reductase (NR); nitrite reductase (NiR) then reduces and transforms NO_2_^−^ into NH_4_^+^ [2]. Lastly, NH_4_^+^ is assimilated into glutamine and glutamate by glutamine synthetase (GS) and glutamate synthase (GOGAT) [4], respectively. However, the N content in soil is limited and mainly depends on the external N supply, such as N fertilizer [5]. The excessive application of N fertilizer leads to environmental pollution, accelerates soil salinization, and reduces the N uptake and use efficiency of plants [6]. Therefore, there is an urgent need to develop methods to reduce the use of N−based fertilizers, to reduce the input cost of fertilizer and protect the environment [7].

Previous studies have shown that the overexpression of *MdBT2* and *OsGRF4* in *Malus hupehensis* and rice promotes N absorption, respectively [8,9]. However, transgenic technology has only been applied to a limited number of crops. Many studies have examined plant hormones, such as cytokinin [7], auxin [10], and ethylene [11], as well as gibberellin (GA). GA is an important endogenous hormone [12], which regulates many key processes in plants, such as the plant stature, axillary meristem outgrowth, leaf development, flowering, and parthenocarpy [13]. GA also regulates the response to low temperature, drought stress, and low light levels [14]. Previous studies have shown that the application of GA together with N fertilizer results in increased crop yields [7]. The effect of GA on N uptake and utilization in plants has received increased research attention. However, the effects of GA on the N absorption and utilization of plants under N−deficient conditions remain unclear.

Pear is the third most economically important fruit in China. However, pear rootstocks face challenges in terms of rooting from cuttings and lack desirable dwarfing characteristics, such as those related to ‘Pyrodwarf (S)’ [15] and ‘Zhong’ai 1′ [16]. ‘Duli’ is the main pear rootstock. By 2021, the cultivated area for pear reached 1,399,484 ha and the global yield was 2,568,713.07 tons [17]. According to the results of this study, we hypothesized that GA_3_ affects the absorption and metabolism of N in ‘Duli’, especially under low NO_3_^−^ conditions. We used a hydroponics system to investigate the role of GA_3_ in plant growth, root architecture, photosynthesis, enzyme activity, in regard to endogenous hormones, and element accumulation. We also explored the expression of genes involved in N metabolism and absorption for two NO_3_^−^ concentrations. Our findings provide new insights into the ability of GA_3_ to reduce the application of N fertilizer in plants.

## 2. Results

### 2.1. Effects of GA_3_ and PAC on the Growth, Chlorophyll Content, and Root Length of ‘Duli’

As shown in Appendix A, the growth of the ‘Duli’ seedlings was inhibited under NO_3_^−^ deficiency; the plant length, leaf number, leaf area per plant, and chlorophyll content were significantly lower, and the main root length was significantly higher under NO_3_^−^ deficiency than in the CK. Exogenous GA_3_ application alleviated the inhibitory effect of NO_3_^−^ deficiency; however, this resulted in a decrease in the chlorophyll content. Paclobutrazol (PAC) had the opposite effect. According to the membership function and its scores (Appendix A), we conclude that the optimal exogenous concentration was 0.1 mM for GA_3_ and 0.01 mM for PAC.

### 2.2. Effects of GA_3_ on the Growth and Chlorophyll Content of ‘Duli’ Seedlings under NO_3_^−^ Deficiency

After 35 days of treatment, the growth of the ‘Duli’ seedlings was weaker under NO_3_^−^ deficiency than in the CK (Figure 1A,B). The plant length, leaf number, leaf area per plant, total fresh weight, and total dry weight, were significantly lower in the SCK and DCK treatments than in the CK (Figure 1C–G); the above parameters were 13.42, 29.27, 7.91, 9.41, and 2.02% higher in the SGK treatment than in the SCK treatment, and 20.90, 32.61, 1.15, 15.42, and 5.89% higher in the DGT treatment than in the DCK treatment, respectively, and these differences were significant (Figure 1C–G). Conversely, the plant length, leaf area per plant, total fresh weight, and total dry weight were lower in the SPT treatment than in the SCK treatment, and lower in the DPT treatment than in the DCK treatment. The root−to−shoot ratio was lower in the GA_3_ treatment than in the same NO_3_^−^ concentration treatment (Figure 1H), which further confirmed that the application of exogenous GA_3_ can promote the growth of ‘Duli’ under NO_3_^−^ deficiency. The variation in the chlorophyll content among the treatments exhibited a similar pattern. The chlorophyll a (Figure 1I), chlorophyll b (Figure 1J), and total chlorophyll content (Figure 1K) was significantly reduced in the SCK and DCK treatments, and the application of GA_3_ had no significant effect on the chlorophyll content. However, after PAC was introduced, the chlorophyll a, chlorophyll b, and total chlorophyll content was higher in the SPT and DPT treatments than in the SCK and DCK treatments. Therefore, PAC could enhance the chlorophyll content, but the application of GA_3_ had no significant effect on the chlorophyll content.

### 2.3. Effects of Exogenous GA_3_ Application on the Root Architecture of ‘Duli’ under NO_3_^−^ Deficiency

The roots are the main organ of N absorption in plants. NO_3_^−^ deficiency significantly promoted root growth (Figure 2A). The main root length (Figure 2B), surface area (Figure 2C), and root tips (Figure 2D) were 29.44, 23.80, and 40.16% higher in the SCK treatment, and 16.76, 12.63, and 23.07% higher in the DCK treatment than in the CK, respectively. The root volume (11.09%) (Figure 2F) was significantly higher in the SCK treatment than in the CK. The root diameter (Figure 2E) was significantly lower in the SCK (20.45%) and DCK (9.22%) treatments than in the CK. The main root length, surface area, root tips, and root volume were 17.72, 5.07, 7.30, and 5.26% higher in the SGT treatment than in the SCK treatment, and 23.08, 6.17, 19.55, and 6.72% higher in the DGT treatment than in the DCK treatment, respectively. The main root length, volume, and tips were significantly lower and the root diameter was higher when PAC was applied at the same NO_3_^−^ concentration. 

### 2.4. Effects of Exogenous GA_3_ on the Photosynthetic Parameters of ‘Duli’ under NO_3_^−^ Deficiency

NO_3_^−^ deficiency had a significant negative effect on the photosynthetic parameters. As shown in Table 1, the *Pn*, *Gs*, *Tr*, *Fo*, *Fv*/*Fm*, and *Rfd* were lower in the SCK and DCK treatments than in the CK. In contrast, the *Ci* and *qP* were higher in the SCK and DCK treatments than in the CK. After GA_3_ application, there were no significant differences in any of the variables between the SGT and SCK treatments, with the exception of *Rfd* and *Ci*. The *Pn* (40.68%), Gs (28.30%), Tr (11.88%), *qP* (11.54%), and *Rfd* (21.48%) were higher in the DGT treatment than in the DCK treatment. When PAC was applied, the *Pn*, *Gs*, *Fo*, and *Fv*/*Fm* significantly increased, but the values of these variables were lower following PAC application than in the CK.

### 2.5. Effects of Exogenous GA_3_ on the Activities of N−Metabolizing Enzymes and Content of Endogenous Hormones in ‘Duli’ under NO_3_^−^ Deficiency

The activities of NR, Fd−GOGAT, and NADH−GOGAT (Figure 3A,C,D) significantly decreased under NO_3_^−^ deficiency in the leaves and roots. The NR and NADH−GOGAT activities were 15.06 and 10.58% higher in the leaves and 23.11 and 34.03% higher in the roots in the SGT treatment than in the SCK treatment, respectively. The activities of GS (5.94%) (Figure 3B) and Fd−GOGAT (14.20%) in the leaves and Fd−GOGAT (9.10%) in the roots was significantly higher in the DGT treatment than in the DCK treatment. In contrast, the activities of N−metabolizing enzymes were lower in the SPT treatment than in the SCK treatment. The activities of NR, GS, Fd−GOGAT, and NADH−GOGAT were lower in the DPT treatment than in the DCK treatment.

The content of GA_3_, IAA, and ZR (Figure 3F,H,I) was similar within the same treatment and the opposite pattern was observed in the ABA content (Figure 3G). The content of GA_3_ and ZR significantly decreased under NO_3_^−^ deficiency. The IAA content significantly decreased under the SCK treatment and it was higher in the DCK treatment than in the CK. The ABA content in the SCK and DCK treatments did not significantly differ from that in the CK. The content of GA_3_, IAA, and ZR was 30.67, 18.63, and 14.82% higher in the SGT treatment than in the SCK treatment, and 41.21, 18.67, and 20.92% higher in the DGT treatment than in the DCK treatment, respectively, and these differences were significant. The ABA content was significantly lower in GA_3_ treatments than the same NO_3_^−^ concentration. By contrast, the exogenous application of PAC resulted in significant decreases in the content of GA_3_, IAA, and ZR. In conclusion, exogenous GA_3_ application promotes the accumulation of growth hormones under NO_3_^−^ deficiency.

### 2.6. Effects of Exogenous GA_3_ Application on the Content of Mineral Elements in ‘Duli’ under NO_3_^−^ Deficiency

The N content of ‘Duli’ seedlings decreased significantly under NO_3_^−^ deficiency (Figure 4A). GA_3_ application increased the N content of the roots, stems, and leaves at the same NO_3_^−^ concentration, and the N content was 6.46, 8.55, and 10.10% higher in the SGT treatment than in the SCK treatment, and 9.08, 8.93, and 11.35% higher in the DGT treatment than in the DCK treatment, respectively. However, the N content was 12.51, 11.95, and 11.35% lower in the SPT treatment than in the SCK treatment, and 7.19, 10.77, and 4.32% lower in the DPT treatment than in the DCK treatment, respectively. The content of the macro− and microelements was affected by NO_3_^−^ deficiency (Figure 4B–J). The content of Ca, Mg, Fe, and Mn in the roots, stems, and leaves was significantly lower in the SCK and DCK treatments than in the CK. The K and Cu content in the leaves decreased significantly under NO_3_^−^ deficiency. In contrast, the P content in ‘Duli’ seedlings was significantly higher in the SCK and DCK treatments than in the CK. Exogenous GA_3_ application significantly increased the K, Ca, Mg, Fe, and Mn content in ‘Duli’ seedlings. However, PAC had no significant effect on the content of most elements at the same NO_3_^−^ concentration.

### 2.7. Effects of Exogenous GA_3_ Application on the Expression of N Uptake and Metabolism−Related Genes in ‘Duli’ under NO_3_^−^ Deficiency 

The key genes involved in N uptake (*NRT2*) and metabolism (*NR*, *NIR*, *NADH−GOGAT*, *Fd−GOGAT*, and *GILE*) were identified using Kyoto Encyclopedia of Genes and Genomes analysis (Appendix A). We also identified a GA signal transduction pathway gene (*GAI1a*). 

The expressions of *PbNRT2* (Figure 5a—A, 5b—A), *PbNR* (Figure 5a—B, 5b—B), *PbNIR* (Figure 5a—C, 5b—C), *PbNADH−GOGAT* (Figure 5a—D, 5b—D), *PbFd−GOGAT* (Figure 5a—E, 5b—E), *PbGILE* (Figure 5a—F, 5b—F), and *PbGAI1a* (Figure 5a—G, 5b—G) were lower under NO_3_^−^ deficiency and PAC application, and the GA_3_ application increased the expression of these genes with the same NO_3_^−^ concentration. The expression of *PbNRT2* was 0.15−and 0.57−fold lower in the leaves and 0.22− and 0.58−fold lower in the roots after 7 days of treatment in the SCK and DCK treatments, respectively, compared with the CK. Exogenous GA_3_ application significantly increased *PbNRT2* expression in the leaves and roots; specifically, the expression of *PbNRT2* was 3.69− and 2.30−fold higher in the SGT treatment than in the SCK treatment and 1.78− and 1.62−fold higher in the DGT treatment than in the DCK treatment, respectively. However, there was no significant difference in *PbNRT2* expression between the DGT and DCK treatments at 14 and 35 days in the leaves. The application of PAC decreased the *PbNRT2* expression. However, *PbNRT2* expression was 2.83−fold higher in the leaves in the SPT treatment than in the SCK treatment. The relative expression of *PbNR*, *PbNIR*, *PbNADH−GOGAT*, *PbFd−GOGAT*, and *PbGILE* was 0.29−, 0.52−, 0.11−, 0.63−, 0.19−, 0.31−, 0.32−, 0.94−, 0.18−, and 0.45−fold lower in the leaves and 0.35−, 0.61−, 0.35−, 0.78−, 0.62−, 0.77−, 0.40−, 0.45−, 0.37−, and 0.70−fold lower in the roots at 35 days, respectively, in the SCK and DCK treatments than in the CK treatment. The relative expression of the above genes (with the exception of *PbFd−GOGAT*) increased at 35 days under the GA_3_ treatment. PAC application decreased the expression of these genes; however, the expression of *PbGILE* was 1.90−fold higher in the leaves in the SPT treatment than in the SCK treatment. The expression of *PbGAI1a* was significantly higher under GA_3_ treatment than under the control conditions; however, no significant differences in the *PbGAI1a* expression in the leaves in the DGT treatment (at 14 days) and SGT treatment (at 28 days), compared with treatments at the same NO_3_^−^ concentration, were observed. PAC application significantly decreased the *PbGAI1a* expression, with the exception of the DPT treatment (at 7 and 14 days) in the leaves and the SPT treatment (at 21 and 35 days) in the roots compared with other treatments at the same NO_3_^−^ concentration. In conclusion, NO_3_^−^ deficiency inhibited the expression of NO_3_^−^ uptake and assimilation genes and *PbGAI1a* in the leaves and roots, and exogenous GA_3_ application promoted the expression of these genes under NO_3_^−^ deficiency.

### 2.8. Correlation and Principal Component Analysis (PCA) 

Correlation coefficients were determined to characterize the correlations between potential indicators and plant length after 35 d of treatment. The plant length was significantly correlated with 29 indicators, according to Pearson correlation analysis (R^2^ > 0.50). In the leaves (Appendix A), the relative expression of *PbNR*, *PbNIR*, *PbFd−GOGAT*, *PbGILE*, and *PbGAI1a*; the content of GA_3_, IAA, ZR, N, K, Ca, Fe, Mg, Mn, and Zn; the NR, GS, Fd−GOGAT, and NADH−GOGAT activity; and the leaf number, total fresh weight, total dry weight, and *Rfd,* were positively correlated with the plant length. In the roots (Appendix A), the relative *PbNR*, *PbNADH−GOGAT*, and *PbFd−GOGAT* expression; the NR, GS, Fd−GOGAT, NADH−GOGAT, and NiR activity; and the Ca, Fe, Mg, and Mn content, were positively correlated with the plant length, and the content of B was negatively correlated with the plant length. This indicates that all these indicators could affect the plant length to varying degrees.

PCA was used to reduce the dimensionality of the 43 indexes (including the aboveground growth and leaf change) to five principal components (PCs) and the total variance explained by these five PCs was 91.42%. The first and second PCs explained 53.74% and 21.72% of the variance, respectively. PC1 was mainly correlated with growth indicators, such as LAP, TFW, TDW, IAA, ZR, NR, N, Ca, Fe, Mn, and *PbNRT2*; PC2 was mainly correlated with the photosynthetic indexes CCA, CCB, CCT, *Pn*, *Gs*, and *Fv*/*Fm* (Figure 6A). The ‘Duli’ samples in the CK were separated along with PC1 from the samples in the SCK and DCK treatments; the other treatments were separated from the CK along with PC2 (Figure 6B). The comprehensive score was highest for the CK, followed by the DGT, DCK, DPT, SGT, SCK, and SPT treatments (Appendix A). This suggests that exogenous GA_3_ application can alleviate the inhibition of the growth of ‘Duli’ under NO_3_^−^ deficiency; however, this inhibitory effect was not completely eliminated.

## 3. Discussion

Plant growth relies on a complex regulatory network and is affected by various factors; one of the most important factors is an adequate supply of essential mineral nutrients, especially N [7,9]. In this study, we found that ‘Duli’ grew (plant length and leaf area per plant) slowly under NO_3_^−^ deficiency compared with the CK, which likely stems from the fact that the N requirements for normal plant growth and development were not met [5]. GA_3_ promoted growth under both of the NO_3_^−^ concentrations applied (0.5 and 8 mM NO_3_^−^). Root architecture growth and structure are important indicators of the ability of plants to absorb N. The roots of ‘Duli’ were larger in the SGT and DGT treatments than in other treatments at the same NO_3_^−^ concentration, as the increase in the root absorption area increased the amount of N absorbed; this has been referred to as the root foraging phenomenon [18]. 

Photosynthesis is an essential physiological process for maintaining the normal growth of plants [19]. In this study, we found that NO_3_^−^ deficiency decreased the rate of photosynthesis and the chlorophyll content. Chlorophyll easily degrades under nutrient stress, which affects the absorption of light energy by leaves and reduces the rate of photosynthesis. However, PAC increased chlorophyll accumulation and promoted photosynthesis; these findings are consistent with the results from previous studies [20]. The precursor of chlorophyll synthesis might be geranylgeranyl pyrophosphate (GGPP), a diterpene associated with the biosynthesis of chlorophyll, and the production of GGPP is inhibited by PAC [21]. The application of PAC might result in the increased conversion of GGPP to diterpenoids, rather than to ketene [22]. Therefore, PAC inhibited chlorophyll degradation under NO_3_^−^ deficiency [23]. The joint regulatory mechanisms of different endogenous hormones play a key role in improving the resistance of plants to N stress [24]. We demonstrated that the application of exogenous GA_3_ led to a significant increase in the endogenous GA_3_ concentration in plants, especially under N stress. Plants can absorb exogenous GA_3_ and accumulate it in their organs, which can have positive effects on plants under N deficiency [25]. Meanwhile, exogenous GA_3_ increases the content of IAA and ZR. The main reason was GA−stimulated IAA production from tryptophan [26]. However, the content of IAA significantly decreased under the SCK treatment. These results are similar to those for rice and *Arabidopsis* [27,28]. This means IAA accumulation is dependent on N [29]. Therefore, the content of IAA in ‘Duli’ was influenced by exogenous GA_3_ and N content in the environment. We also found that the application of exogenous GA_3_ promotes photosynthesis and no significant difference in the chlorophyll content was observed among the treatments under the same NO_3_^−^ concentration. The main reason was that exogenous GA_3_ application decreases the content of endogenous hormones, such as ABA, and inhibits plant stomatal closure to improve the photosynthetic capacity [19,30].

The role of N assimilation in plant growth and development has been extensively studied by researchers [31]. Excessive or insufficient N reduces the activity of N metabolism−related enzymes [5]. Previous studies have shown that the application of GA_3_ increased the activity of enzymes related to N metabolism in plants [32]. NR plays a key role in NO_3_^−^ assimilation [11]. In this study, we found that NR enzyme activity was increased under a certain NO_3_^−^ concentration range. Additionally, the activities of NR and NiR are affected by GA_3_, and this has also been reported in regard to tobacco [33]. However, in this study, the activity of NiR only increased in the SGT treatment, and NR activity was affected by the SGT and DGT treatments, indicating that NR was sensitive to GA_3_.

The adequate absorption of minerals is important for the maintenance of plant structural integrity and key physiological processes [34]. We found that the N and Fe content in ‘Duli’ seedlings decreased significantly under NO_3_^−^ deficiency, which might stem from an interaction between elements [35]. GA_3_ increased the N content, and this has also been reported in cucumber [36] and rice [8]. Ionomics analysis revealed that the application of exogenous GA_3_ promotes the absorption of P, K, Ca, and Mg. This is because GA_3_ regulates the expression genes that regulate the absorption of various elements, such as *SlPT2* and *SlPT7* (Pi transporter) [37], *AtHAK5* (K transporter) [38], and *AtIRT1* (Fe−regulated transporter) [39]. 

*NRT2* is a high NO_3_^−^ affinity family member [40]. A previous study has reported that *NRT2* transcription levels are affected by GA_3_ and NO_3_^−^ [41]. We found that the expression of *PbNRT2* increased under the increased NO_3_^−^ concentration and under exogenous GA_3_ application; similar findings have been reported in maize [42]. We also found that GA_3_ enhanced the expression of *PbNR*, *PbNiR*, *PbGS*, *PbNADH−GOGAT*, and *PbFd−GOGAT* under NO_3_^−^ deficiency to promote the absorption and transport of NO_3_^−^ in the leaves and roots of ‘Duli’ seedlings; similar findings have been obtained in *Arabidopsis* [43]. The expression levels of *PbGAI1a* (which encodes the DELLA protein) significantly increased under GA_3_ application, compared with other treatments in which GA_3_ was not applied at the same NO_3_^−^ concentration. Exogenous GA_3_ application increases the endogenous GA_3_ content, which increases the expression of *PbGAI1a* [44]. Our previous study confirmed that the overexpression of *PbGAI1a* effectively reduces the plant length of *Arabidopsis*. However, the overexpression of DELLA significantly reduces N utilization [8]. In rice, the DELLA−GRF4 (growth regulator factor 4) model clarifies the relationship between dwarfing and N utilization [8]. Therefore, future studies are needed to explore the other mechanism of DELLA−N in pears.

## 4. Materials and Methods

### 4.1. Plant Materials and Growth Conditions

All the experiments were performed at Innovation Pilot Park, Hebei Agricultural University (38.23° N, 115.28° E). Seeds of ‘Duli’ were collected from Zanhuang (37.67° N. 114.38° E) in Hebei, China. After 30 days of storage at 4 °C for stratification, the seeds were planted into plastic containers filled with sand for seed germination and growth. One month later, the seedlings (similar size, with 5–6 leaves and 6 cm in height) were transplanted into gray hydroponic pots to be pre−cultured for 1 week (with 10 L 1/2 strength Hoagland nutrient solution, pH: 6.5 ± 0.1). The plantlets were grown under a 14 h/10 h light (23–25 °C)/dark (19–21 °C) photoperiod, with a relative humidity of 60–80% and light intensity of 37.04 μM m^-2^ s^-1^, the oxygen content was maintained via air pumps, and the solution was replaced once a week [45].

In this study, Ca(NO_3_)_2_ (Sinopharm Chemical Reagent Co., Ltd., Shanghai, China) was the only N source. The optimal concentration of NO_3_^−^ in the hydroponic system for ‘Duli’ is 16 mM and this concentration was maintained to ensure normal plant growth [46]. After pre−culturing for 1 week, 550 healthy seedlings were selected and divided into eleven experimental groups (each group containing 50 seedlings), including the CK, with 16 mM NO_3_^−^ Hogland nutrition solution, and exogenous GA_3_ (0, 0.01, 0.05, 0.1, and 0.15 mM) (BBI, Shanghai, China) and PAC (0.005, 0.01, 0.02, 0.04, and 0.1 mM) (BBI, Shanghai, China) with 0.5 mM NO_3_^−^. 

In addition, 1050 healthy seedlings were selected and divided into seven experimental groups according to the NO_3_^−^ concentration and exogenous substances: (1) SCK, 0.5 mM NO_3_^−^; (2) SGT, 0.5 mM NO_3_^−^ with 0.1 mM GA_3_; (3) SPT, 0.5 mM NO_3_^−^ with 0.01 mM PAC; (4) DCK, 8 mM NO_3_^−^; (5) DGT, 8 mM NO_3_^−^ with 0.1 mM GA_3_; (6) DPT, 8 mM NO_3_^−^ with 0.01 mM PAC; and (7) CK, 16 mM NO_3_^−^ (Appendix A).

### 4.2. Analysis of Growth, Chlorophyll Content, and Root System Architecture

The plant length, leaf number, leaf area per plant, total fresh weight, total dry weight, and root−to−shoot ratio, were determined following the methods by Du et al. [5]. Ten healthy seedlings were selected; each plant was divided into roots, stems, and leaves; washed with deionized water and dried with a paper towel. The plants were fixed at 105 °C for 30 min, then dried at 65 °C to a constant weight, and the total dry weight was evaluated by an electronic balance. Ten replications of each treatment were performed. The chlorophyll a, chlorophyll b, and total chlorophyll content was determined using a UV−1800 spectrophotometer (UV−1800, Metash, Shanghai, China) [12,47]. The leaves (0.1 g with the main vein removed) were obtained and extraction was performed with 10 mL of 80% acetone for more than 24 h in the dark. The absorbance of the extract was measured at wavelengths of 663 and 645 nm. Five replications of each treatment were performed. The roots were imaged using an Epson digital scanner and analyzed using the WinRHIZO^®^ image analysis system (V4.1c; Régent Instruments, Quebec, Canada) [5]. The main root length was measured using a scaled ruler. Ten replications of each treatment were performed.

### 4.3. Photosynthetic Parameters and Chlorophyll Fluorescence Determination

The third to fifth mature and fully exposed leaves from the top of the plants were used to determine the photosynthetic parameters; five replications of each treatment were performed. On sunny days, between 09:00 and 11:00 h, the net photosynthetic rate (*Pn*), stomatal conductance (*Gs*), intercellular CO_2_ concentration (*Ci*), and transpiration rate (*Tr*), were monitored with a Li−Cor portable photosynthesis system (Li6400; LICOR, Huntington Beach, CA, USA). The photosynthetic readings were taken at 1000 μM photons m^−2^ s^−1^ and a constant airflow rate of 500 μM s^−1^. The cuvette CO_2_ concentration was set to 400 μM CO_2_ mol^−1^ air [48]. The minimal fluorescence (*Fo*), maximum photochemical efficiency of PSII (*Fv*/*Fm*), photochemical quenching (*qP*), and steady−state fluorescence decay rate (*Rfd*) of the functional leaves were determined using a portable pulse−modulated fluorometer (Hansatech, Norfolk, Virginia, UK) [49].

### 4.4. N-Metabolizing Enzyme Activities and Endogenous Hormone Measurements

The NR, NiR, glutamine synthetase (GS), ferredoxin−dependent glutamate synthase (Fd−GOGAT), and nicotinamide adenine dinucleotide (NADH−GOGAT) activities in the leaves and roots were determined using relevant kits (Geruisi, Suzhou, China). The content of GA_3_, indole−3−acetic acid (IAA), zeatin riboside (ZR), and abscisic acid (ABA) was determined using high−performance liquid chromatography (HPLC, LC−2010, Shimazu, Japan). The extraction and determination methods were based on those described in our previous study [50]. Three replications of each treatment were performed.

### 4.5. Elemental Measurements

After 35 days of treatment, the seedlings were divided into roots, stems, and leaves, and washed with 1% (*w*/*v*) citric acid and deionized water, twice. After fixing at 105 °C for 15 min and being oven−dried at 70 °C to a constant weight, the samples were ground, mixed, and sieved. Finally, a 0.1 g sample of ash was digested with 10 mL HNO_3_, using a microwave digestion system (MARS, CEM Corporation, Matthews, NC, USA). 

The N and phosphorus (P) content was determined using a continuous flow analyzer (Auto Analyzer 3, SEAL Analytical, Norderstedt, Germany). The potassium (K), calcium (Ca), magnesium (Mg), iron (Fe), manganese (Mn), zinc (Zn), boron (B), and copper (Cu) content was determined using inductively coupled plasma source mass spectrometry (ICP, Thermo Fisher Scientific Co., Waltham, MA, USA) [1]. Three replications of each treatment were performed.

### 4.6. qRT-PCR Analysis

The total RNA extraction and reverse transcription of ‘Duli’ leaves and roots were conducted following the methods by Song et al. [44]. Premier 5.0 (Premier Biosoft International, Silicon Valley, CA, USA) was used to design the primers for the qPCR and *PbActin* was used as the internal reference standard (Appendix A). The 2^−∆∆CT^ method was used to calculate the relative expression level of the genes.

### 4.7. Statistical Analysis 

The membership function method was used to calculate the membership function values of the ‘Duli’ seedling growth, chlorophyll content, and main root length under different GA_3_/PAC concentrations. The formulas are as follows: R(Xi) = (Xi − Xmin)/(Xmax − Xmin)(1)
R(Xi) = 1 − (Xi − Xmin)/(Xmax − Xmin)(2)
where Xi represents the measured value of a specific index; Xmax and Xmin represent the maximum and minimum values of that index among all materials, respectively; and R(Xi) represents the membership degree value. Formula (1) was used as the index because it is positively correlated with growth, and Formula (2) was used as the index because it is negatively correlated with growth [51]. All the experimental data were analyzed using a one−way ANOVA, followed by Tukey’s test (*p* < 0.05), to determine the significance of the differences between treatments. SPSS 25 (SPSS Inc., Chicago, IL, USA) and Origin 2019 (Origin Lab, Northampton, MA, USA) were used to analyze the experimental data. The data were expressed as mean ± standard deviation.

## 5. Conclusions

The experimental results showed that the aboveground growth of ‘Duli’ was significantly inhibited under low NO_3_^−^ conditions. The application of GA_3_ increased the expression of genes related to N uptake and metabolism and the activity of N−metabolizing enzymes, regulated the accumulation of N and other elements, and alleviated the inhibitory effect of NO_3_^−^ deficiency on ‘Duli’ growth (Figure 7). We suggest that the positive effect of GA_3_ could be leveraged to promote the growth of plants under N−deficient conditions and enhance adaptation to future environmental challenges.

## Figures and Tables

**Figure 1 ijms-25-07967-f001:**
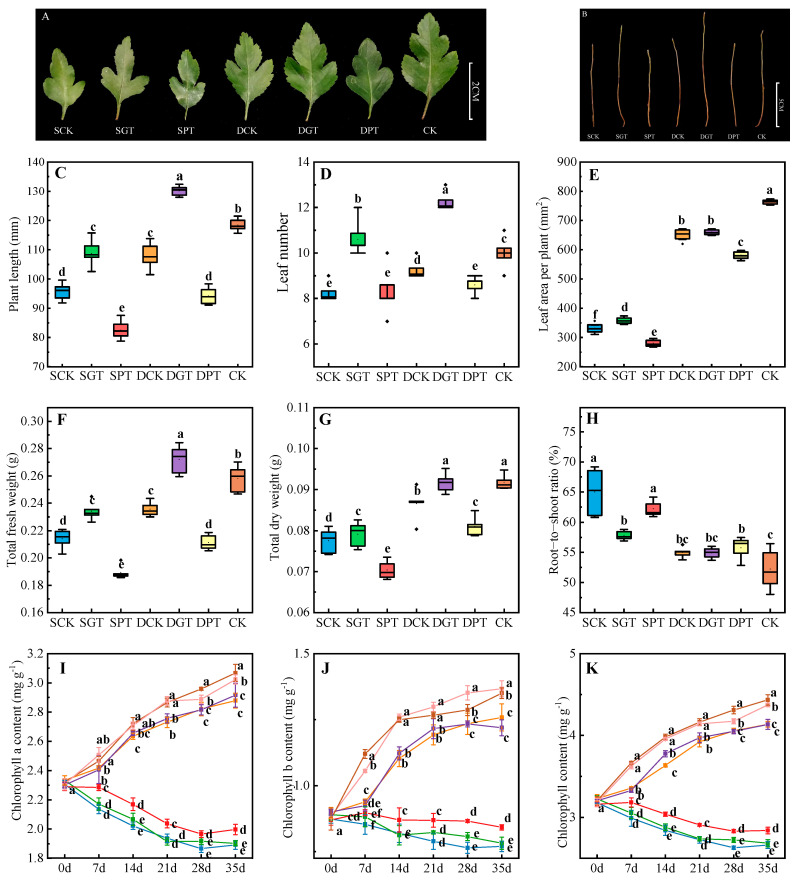
Effects of GA_3_ on the growth and chlorophyll content of ‘Duli’ seedlings under NO_3_^−^ deficiency. Data are presented as means ± SD (the growth *n* = 10 and chlorophyll content *n* = 5). (**A**), leaf phenotype; (**B**), stem phenotype; (**C**), plant length; (**D**), leaf number; (**E**), leaf area per plant; (**F**), total fresh weight; (**G**), total dry weight; (**H**), root−to−shoot ratio; (**I**), chlorophyll a content; (**J**), chlorophyll b content; (**K**), chlorophyll content. Values not followed by the same letter denote significant differences based on Tukey’s multiple−range tests (*p* < 0.05). SCK, 0.5 mM NO_3_^−^ solution; SGT, 0.5 mM NO_3_^−^ solution with 0.1 mM GA_3_; SPT, 0.5 mM NO_3_^−^ solution with 0.01 mM PAC; DCK, 8 mM NO_3_^−^ solution; DGT, 8 mM NO_3_^−^ solution with 0.1 mM GA_3_; DPT, 8 mM NO_3_^−^ solution with 0.01 mM PAC; CK, 16 mM NO_3_^−^ solution.

**Figure 2 ijms-25-07967-f002:**
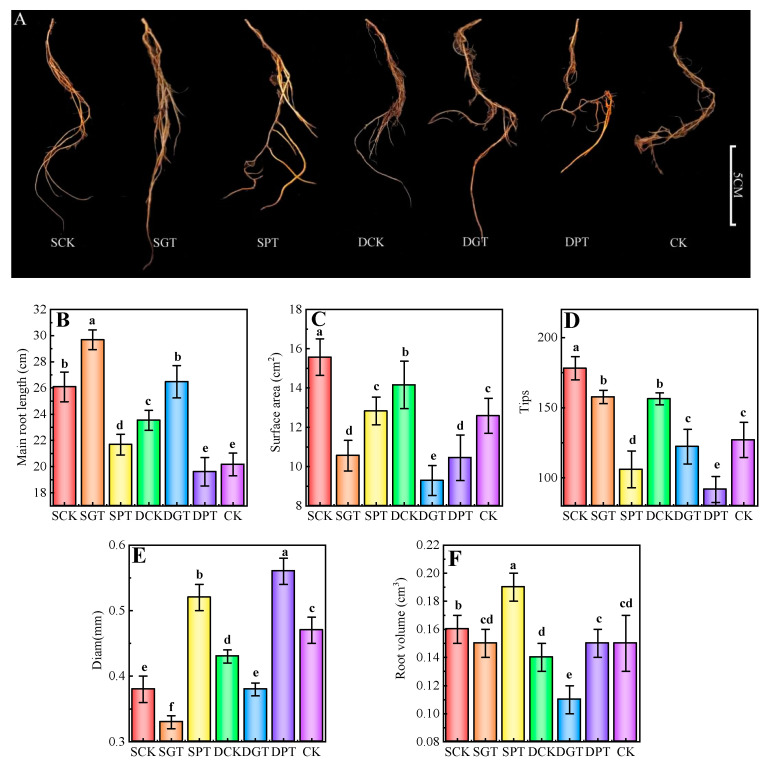
Effects of exogenous GA_3_ on the root architecture of ‘Duli’ seedlings under NO_3_^−^ deficiency. Data are presented as means ± SD (*n* = 10). (**A**), root phenotype; (**B**), main root length; (**C**), surface area; (**D**), tips; (**E**), diam; (**F**), root volume. Values not followed by the same letter denote significant differences based on Tukey’s multiple−range tests (*p* < 0.05). SCK, 0.5 mM NO_3_^−^ solution; SGT, 0.5 mM NO_3_^−^ solution with 0.1 mM GA_3_; SPT, 0.5 mM NO_3_^−^ solution with 0.01 mM PAC; DCK, 8 mM NO_3_^−^ solution; DGT, 8 mM NO_3_^−^ solution with 0.1 mM GA_3_; DPT, 8 mM NO_3_^−^ solution with 0.01 mM PAC; CK, 16 mM NO_3_^−^ solution.

**Figure 3 ijms-25-07967-f003:**
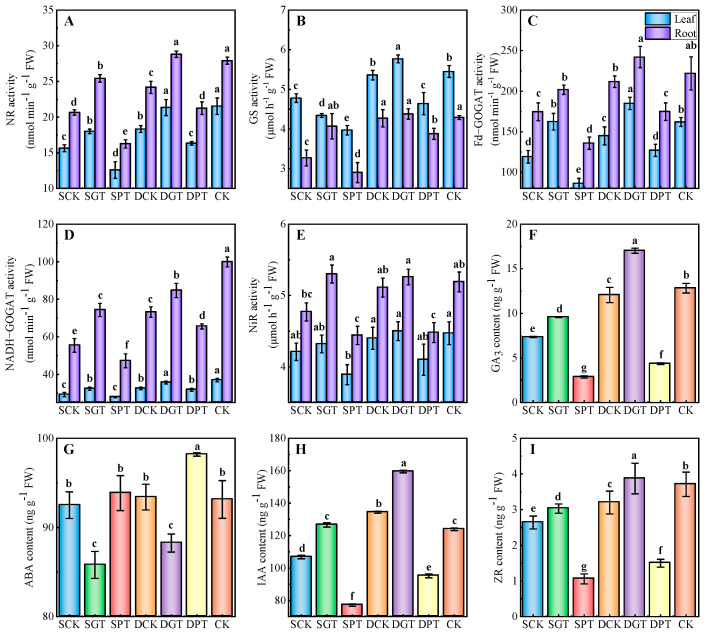
Effects of exogenous GA_3_ on the N−metabolizing enzyme activities and endogenous hormone content of ‘Duli’ seedlings under NO_3_^−^ deficiency. (**A**), nitrate reductase (NR); (**B**), glutamine synthetase (GS); (**C**), ferredoxin−dependent glutamate synthase (Fd−GOGAT); (**D**), nicotinamide adenine dinucleotide (NADH−GOGAT); (**E**), nitrite reductase (NiR); (**F**), gibberellin acid (GA_3_); (**G**), indole−3−acetic acid (IAA); (**H**), zeatin riboside (ZR); (**I**), abscisic acid (ABA). Data are presented as means ± SD (*n* = 3). Values not followed by the same letter denote significant differences based on Tukey’s multiple−range tests (*p* < 0.05). SCK, 0.5 mM NO_3_^−^ solution; SGT, 0.5 mM NO_3_^−^ solution with 0.1 mM GA_3_; SPT, 0.5 mM NO_3_^−^ solution with 0.01 mM PAC; DCK, 8 mM NO_3_^−^ solution; DGT, 8 mM NO_3_^−^ solution with 0.1 mM GA_3_; DPT, 8 mM NO_3_^−^ solution with 0.01 mM PAC; CK, 16 mM NO_3_^−^ solution.

**Figure 4 ijms-25-07967-f004:**
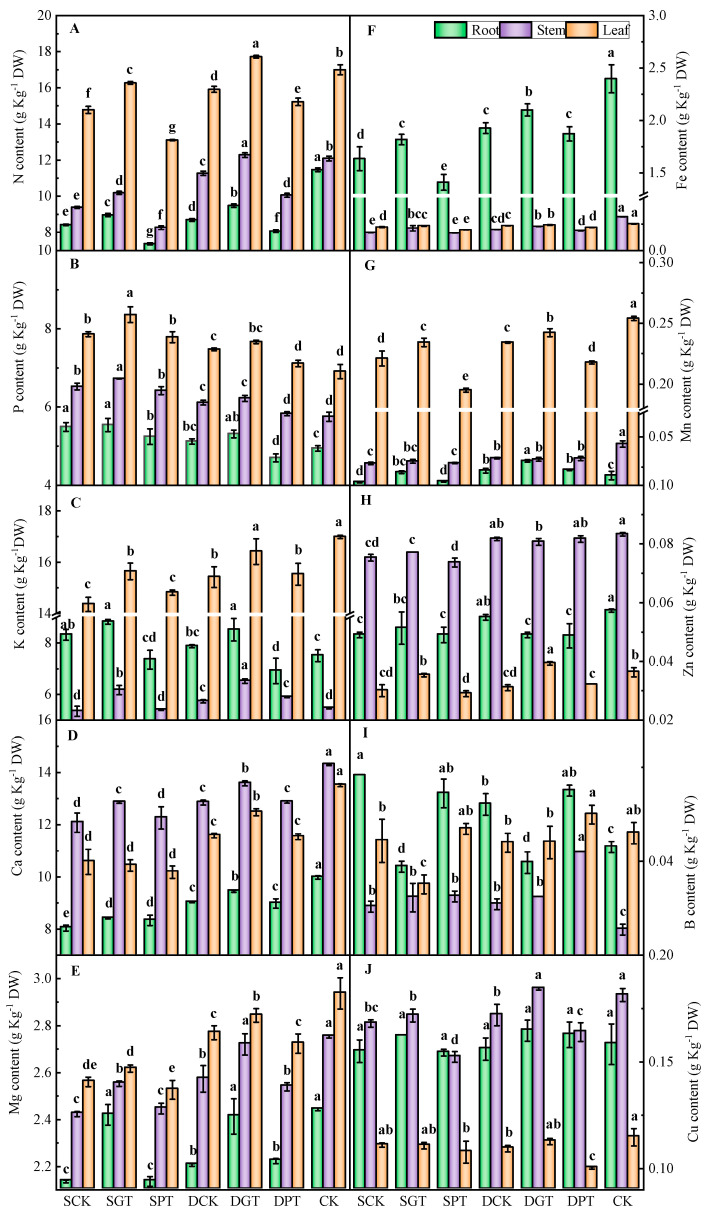
Effects of exogenous GA_3_ on the mineral element content of ‘Duli’ seedlings under NO_3_^−^ deficiency. Data are presented as means ± SD (*n* = 3). (**A**), nitrogen (N); (**B**), phosphorus (P); (**C**), potassium (K); (**D**), calcium (Ca); (**E**), magnesium (Mg); (**F**), iron (Fe); (**G**), manganese (Mn); (**H**), zinc (Zn); (**I**), boron (B); (**J**), cuprum (Cu). Values not followed by the same letter denote significant differences based on Tukey’s multiple−range tests (*p* < 0.05). SCK, 0.5 mM NO_3_^−^ solution; SGT, 0.5 mM NO_3_^−^ solution with 0.1 mM GA_3_; SPT, 0.5 mM NO_3_^−^ solution with 0.01 mM PAC; DCK, 8 mM NO_3_^−^ solution; DGT, 8 mM NO_3_^−^ solution with 0.1 mM GA_3_; DPT, 8 mM NO_3_^−^ solution with 0.01 mM PAC; CK, 16 mM NO_3_^−^ solution.

**Figure 5 ijms-25-07967-f005:**
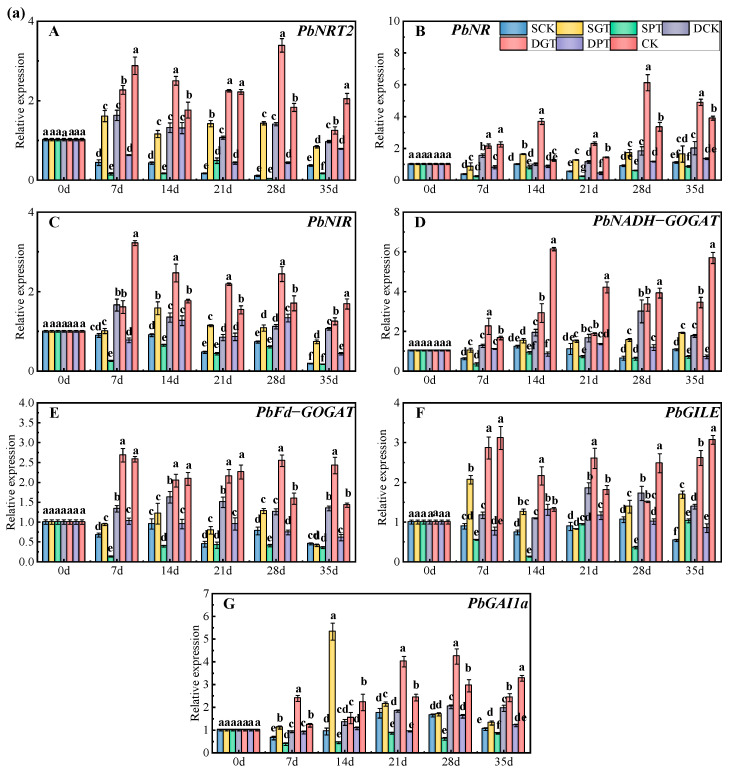
Effects of exogenous GA_3_ on the expression of key genes involved in N uptake and metabolism of ‘Duli’ under NO_3_^−^ deficiency: (**a**), leaves. (**b**), roots. Data are presented as means ± SD (*n* = 3). Values not followed by the same letter denote significant differences based on Tukey’s multiple−range tests (*p* < 0.05). SCK, 0.5 mM NO_3_^−^ solution; SGT, 0.5 mM NO_3_^−^ solution with 0.1 mM GA_3_; SPT, 0.5 mM NO_3_^−^ solution with 0.01 mM PAC; DCK, 8 mM NO_3_^−^ solution; DGT, 8 mM NO_3_^−^ solution with 0.1 mM GA_3_; DPT, 8 mM NO_3_^−^ solution with 0.01 mM PAC; CK, 16 mM NO_3_^−^ solution.

**Figure 6 ijms-25-07967-f006:**
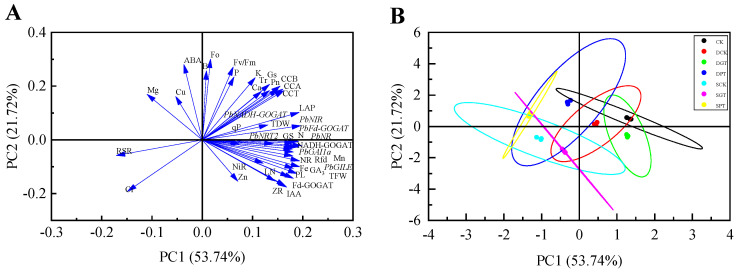
Principal component analysis based on 43 characters. (**A**), loading plot of 43 variables in the factor plane; (**B**), component graph in two-dimensional rotated space. The plant length (PL); leaf number (LN); leaf area per plant (LAP); total fresh weight (TFW); total dry weight (TDW); root−to−shoot ratio (RSR); chlorophyll a (CCA); chlorophyll b (CCB); total chlorophyll content (CCT); net photosynthetic rate (*Pn*); stomatal conductance (*Gs*); intercellular CO_2_ concentration (*Ci*); transpiration rate (*Tr*); minimal fluorescence (*Fo*); maximum photochemical efficiency of PSII (*Fv*/*Fm*); photochemical quenching (*qP*); steady−state fluorescence decay rate (*Rfd*); nitrate reductase (NR); glutamine synthetase (GS); ferredoxin−dependent glutamate synthase (Fd−GOGAT); nicotinamide adenine dinucleotide (NADH−GOGAT); nitrite reductase (NiR); gibberellin acid (GA_3_); indole−3−acetic acid (IAA); zeatin riboside (ZR); abscisic acid (ABA); nitrogen (N); phosphorus (P); potassium (K); calcium (Ca); magnesium (Mg); iron (Fe); manganese (Mn); zinc (Zn); boron (B); cuprum (Cu). SCK, 0.5 mM NO_3_^−^ solution; SGT, 0.5 mM NO_3_^−^ solution with 0.1 mM GA_3_; SPT, 0.5 mM NO_3_^−^ solution with 0.01 mM PAC; DCK, 8 mM NO_3_^−^ solution; DGT, 8 mM NO_3_^−^ solution with 0.1 mM GA_3_; DPT, 8 mM NO_3_^−^ solution with 0.01 mM PAC; CK, 16 mM NO_3_^−^ solution.

**Figure 7 ijms-25-07967-f007:**
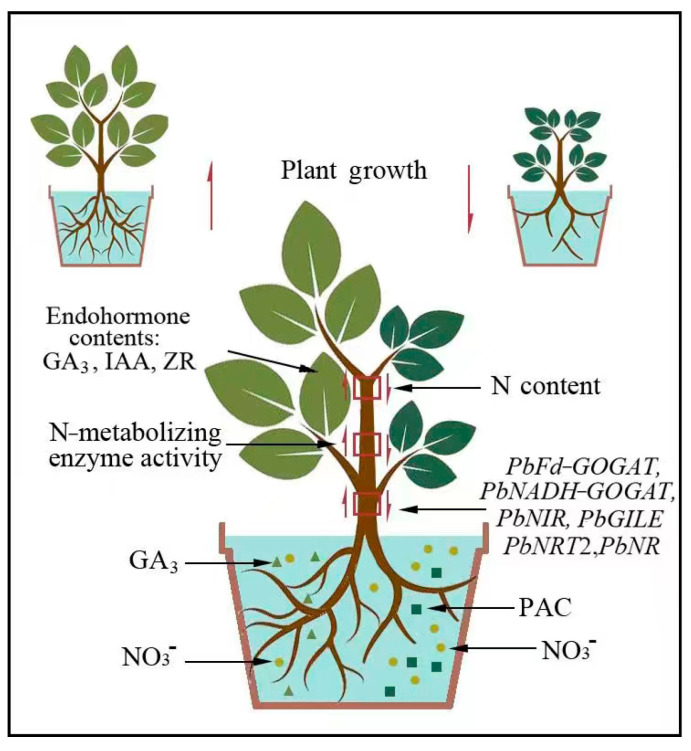
Proposed model of the potential mechanisms by which GA_3_ promotes N uptake and metabolism in ‘Duli’. Under NO_3_^−^ deficiency, the application of GA_3_ increases the endogenous hormone GA_3_ content, the activity of N−metabolizing enzymes, induces the accumulation of N, and increases the expression of N absorption and assimilation genes.

**Table 1 ijms-25-07967-t001:** Effects of exogenous GA_3_ on the photosynthetic parameters of ‘Duli’ seedlings under NO_3_^−^ deficiency. The net photosynthetic rate (*Pn*); stomatal conductance (*Gs*); intercellular CO_2_ concentration (*Ci*); transpiration rate (*Tr*); minimal fluorescence (*Fo*); maximum photochemical efficiency of PSII (*Fv*/*Fm*); photochemical quenching (*qP*); steady−state fluorescence decay rate (*Rfd*). Data are presented as means ± SD (*n* = 5). Values not followed by the same letter denote significant differences based on Tukey’s multiple−range tests (*p* < 0.05). SCK, 0.5 mM NO_3_^−^ solution; SGT, 0.5 mM NO_3_^−^ solution with 0.1 mM GA_3_; SPT, 0.5 mM NO_3_^−^ solution with 0.01 mM PAC; DCK, 8 mM NO_3_^−^ solution; DGT, 8 mM NO_3_^−^ solution with 0.1 mM GA_3_; DPT, 8 mM NO_3_^−^ solution with 0.01 mM PAC; CK, 16 mM NO_3_^-^ solution.

Treatments	*Pn* (µM CO_2_ m^−2^ s^−1^)	*Gs* (mol H_2_O m^−2^ s^−1^)	*Ci* (µM CO_2_ m^−2^ s^−1^)	*Tr* (mM H_2_O m^−2^ s^−1^)	*Fo*	*Fv*/*Fm*	*qP*	*Rfd*
SCK	12.02 ± 0.30 e	0.30 ± 0.02 e	371.24 ± 13.86 a	11.61 ± 0.67 e	120.41 ± 4.83 d	0.77 ± 0.05 c	0.25 ± 0.04 b	1.44 ± 0.03 e
SGT	12.97 ± 0.55 e	0.28 ± 0.01 e	341.08 ± 19.95 b	11.89 ± 0.40 e	118.05 ± 7.32 d	0.77 ± 0.04 c	0.27 ± 0.03 ab	1.55 ± 0.03 c
SPT	14.80 ± 0.07 d	0.50 ± 0.02 d	284.46 ± 9.42 c	16.48 ± 0.42 d	159.06 ± 5.30 ab	0.83 ± 0.02 b	0.27 ± 0.01 ab	1.43 ± 0.02 e
DCK	15.78 ± 0.23 c	0.53 ± 0.04 c	281.65 ± 9.68 c	17.17 ± 0.48 bc	136.29 ± 6.38 c	0.83 ± 0.01 b	0.26 ± 0.01 ab	1.49 ± 0.02 d
DGT	16.91 ± 0.36 a	0.68 ± 0.01 a	246.11 ± 21.78 d	19.21 ± 0.12 a	139.61 ± 4.69 c	0.82 ± 0.01 b	0.29 ± 0.01 a	1.81 ± 0.01 a
DPT	16.25 ± 0.39 b	0.62 ± 0.01 b	258.16 ± 19.51 cd	16.76 ± 1.32 cd	166.76 ± 3.26 a	0.89 ± 0.02 a	0.27 ± 0.02 ab	1.50 ± 0.03 d
CK	16.54 ± 0.05 ab	0.60 ± 0.01 b	201.99 ± 10.54 e	17.48 ± 1.77 b	151.44 ± 5.20 b	0.86 ± 0.01 a	0.27 ± 0.01 ab	1.69 ± 0.02 b

## Data Availability

Data are contained within the article and Appendix A.

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
