# Peer review of "Exogenous GA3 Enhances Nitrogen Uptake and Metabolism under Low Nitrate Conditions in ‘Duli’ (Pyrus betulifolia Bunge) Seedlings"

_ijms, 2024, doi:10.3390/ijms25147967_

Round 1
Reviewer 1 Report
Comments and Suggestions for Authors'Duli' (Pyrus betulifolia Bunge.) is a widely used rootstock for pear trees in northern China, known for its tall stature and well-developed taproot system, but it faces challenges in rooting from cuttings and lacks desirable dwarfing characteristics. A study investigating the effects of exogenous gibberellic acid (GA3) on 'Duli' seedlings under nitrate deficiency revealed that GA3 significantly improves growth by altering root architecture, enhancing hormone content, boosting photosynthesis, and increasing nitrogen metabolism and uptake. Overall, this is a good-quality article and the results could be useful in horticultural production. I only have a few minor suggestions that should be considered before publishing the manuscript.
- Add initials of the species in the title and whenever mentioning a species for the first time, regardless of the Abstract (e.g. Pyrus betulifolia Bunge)
- Names of cultivars should not be written in italics
- Keywords should not repeat words from the title
- Add details on the producers of key chemicals and equipment used (name city, state, and country).
- Lux is an outdated, non-informative unit. What was the PPFD value?
- How did you obtain the dry weight?
- The numbering of Figures is incorrect (e.g. There is no Figure 3, 8 or 9).
- Not all relevant recent literature related to the propagation of Pear rootstock is cited.
- Doi numbers are missing in the Reference list
Comments on the Quality of English LanguageEnglish is OK
Author Response
Comments 1:
Question 1: Add initials of the species in the title and whenever mentioning a species for the first time, regardless of the Abstract (e.g. Pyrus betulifolia Bunge).
Response: Thank you for pointing this out. We agree with this comment. Therefore, we have added it (Line 3, 8).
Question 2: Names of cultivars should not be written in italics
Response: Thank you for pointing this out. We agree with this comment. Therefore, we have revised it in this manuscript.
Question 3: Keywords should not repeat words from the title.
Response: Thank you for pointing this out. We agree with this comment. Therefore, we have revised it as ‘plant endogenous hormone; pear rootstocks’ (Line 19-20).
Question 4: Add details on the producers of key chemicals and equipment used (name city, state, and country).
Response: Thank you for pointing this out. We agree with this comment. Therefore, we have added it. Ca(NO3)2 (Sinopharm Chemical Reagent Co., Ltd, Shanghai, China) (Line 361); GA3 (BBI, Shanghai, China) and PAC (BBI, Shanghai, China) (Line 336-337); UV-1800 spectrophotometer (UV-1800, Metash, Shanghai, China) (Line 382); Li-Cor portable photosynthesis system (Li6400; LICOR, Huntington Beach, CA, USA) (Line 394-395); high-performance liquid chromatography (HPLC, LC-2010, Shimazu, Japan) (Line 406-407).
Question 5: Lux is an outdated, non-informative unit. What was the PPFD value?
Response: Thank you for pointing this out. We agree with this comment. Therefore, we have revised it. The PPFD value was 37.04 μM/m2/s (Line 359).
Question 6: How did you obtain the dry weight?
Response: Thank you for pointing this out. We have added ‘Ten healthy seedlings were selected, each plant was divided into roots, stems, and leaves; washed with deionized water and dried with a paper towel. The plants were fixed at 105 ℃ for 30 min, then dried at 65 ℃ to constant weight, and total dry weight was evaluated by an electronic balance’ (Line 376-380).
Question 7: The numbering of Figures is incorrect (e.g. There is no Figure 3, 8 or 9).
Response: Thank you for pointing this out. We agree with this comment. Therefore, we have revised it.
Question 8: Not all relevant recent literature related to the propagation of Pear rootstock is cited.
响应: 谢谢你指出这一点。我们同意这一评论。因此,我们添加了它。“然而,梨砧木在扦插生根方面面临挑战,缺乏理想的矮化特征,如'Pyrodwarf(S)'和'Zhong'ai 1'。”(第 49-51 行)。
问题 9: 参考文献列表中缺少 Doi 号
响应:谢谢你指出这一点。我们同意这一评论。因此,我们添加了它(第 475-605 行)。

Reviewer 2 Report
Comments and Suggestions for Authors
Review of the Article “Exogenous GA3 Enhances Nitrogen Uptake and Metabolism Under Low Nitrate Levels in ‘duli’ (Pyrus betulifolia) Seedlings”
Objective:
The objective of the paper is well-defined and addresses the effect of exogenous GA3 on nitrogen uptake and metabolism of 'duli' seedlings under nitrate-deficient conditions. The study aims to investigate how GA3 affects growth, root architecture, photosynthesis, enzyme activity, endogenous hormone levels and gene expression related to nitrogen metabolism. This is clearly stated in the introduction and supported by a comprehensive set of experiments.
Structure:
Abstract: The abstract effectively summarizes the aims, methods, results and conclusions of the study. It provides a clear and concise overview of the study.
Introduction: The introduction is thorough and provides context about the importance of nitrogen in plant growth, the role of gibberellins and the specific relevance to 'duli' seedlings. It appropriately sets the stage for the research by citing relevant literature and identifying gaps that the current study aims to fill.
Materials and methods: This section is detailed and well-organized. It includes specific information on plant materials, growth conditions, experimental design, measurement techniques and statistical analyses. However, some minor improvements could be made:
Clarify the selection criteria for healthy seedlings.
Include a flow chart or diagram to illustrate the experimental design.
Results: The results are comprehensively presented with clear subsections detailing the effects of GA3 and PAC on various growth parameters, photosynthesis, enzyme activities, hormone levels and gene expression.
The data are well supported by tables, figures and statistical analyses.
Figures and tables are appropriately labelled and referenced in the text.
Supplementary data are used effectively to provide additional insight.
Discussion: The discussion effectively interprets the results and relates them to the aims of the study and the existing literature.
It explains the significance of the results and their implications for nitrogen metabolism in 'duli' seedlings.
The potential mechanisms by which GA3 enhances nitrogen uptake are discussed.
Limitations of the study are acknowledged and suggestions for future research are provided.
Conclusion: The conclusion succinctly summarizes the main findings and their practical implications. It reinforces the potential benefits of GA3 in promoting plant growth under nitrogen-deficient conditions.
References: References are appropriate and up-to-date, covering a wide range of relevant studies.
Strengths:
Comprehensive experimental design: The study uses a robust experimental design with multiple treatments and replicates, which ensures the reliability of the results.
Detailed data presentation: Results are presented in a clear and detailed manner, supported by well-designed figures and tables.
Thorough discussion: The discussion effectively integrates the findings of the study with existing knowledge, providing a well-rounded interpretation of the results.
Need to improve:
Methodological clarity: Although the methods section is detailed, including a flowchart or schematic of the experimental setup would improve clarity and help readers better understand the experimental design.
Expanded discussion of hormones: The discussion could benefit from a more detailed analysis of the interaction between different endogenous hormones and their collective effect on nitrogen metabolism.
Future research directions: Although the discussion includes suggestions for future research, it would be beneficial to identify specific areas or questions for further investigation.
The article meets the requirements of a scientific research paper. The aim is well stated, the experimental design is robust, and the results are presented and discussed. Minor improvements in methodological clarity and expanded discussion of hormone interactions would improve the overall quality of the paper.
Overall, the article is well-written, and comprehensive and provides valuable insights into the role of GA3 in nitrogen metabolism under nitrate-deficient conditions in 'duli' seedlings. With minor adjustments, it can achieve a higher level of clarity and impact.
Author Response
Question 1: Clarify the selection criteria for healthy seedlings.
Response: Thank you for pointing this out. We agree with this comment. Therefore, we have added it. Similar size, with 5–6 leaves and 6 cm in height (Line 355).
Question 2: Methodological clarity: Although the methods section is detailed, including a flowchart or schematic of the experimental setup would improve clarity and help readers better understand the experimental design?
Response: Thank you for pointing this out. We agree with this comment. We have added the experimental design table in the supplementary Table S4.
Question 3: Expanded discussion of hormones: The discussion could benefit from a more detailed analysis of the interaction between different endogenous hormones and their collective effect on nitrogen metabolism.
Response: Thank you for pointing this out. We agree with this comment. Therefore, we have revised it. Meanwhile, exogenous GA3 increases the content of IAA and ZR. The main reason was GA-stimulated IAA production from tryptophan. However, the content of IAA significantly decreased under the SCK treatment. These results are similar to rice and Arabidopsis. This means IAA accumulation is dependent on N. Therefore, the content of IAA in ‘duli’ was influenced by exogenous GA3 and N content in the environment (Line 305-310).
Question 4: Future research directions: Although the discussion includes suggestions for future research, it would be beneficial to identify specific areas or questions for further investigation.
Response: Thank you for pointing this out. We agree with this comment. Therefore, we have revised it. Our previous study has confirmed that overexpression of PbGAI1a effectively reduces the plant length of Arabidopsis. However, overexpression of DELLA significantly reduces N utilization. In rice, the DELLA-GRF4 (growth regulator factor 4) model clarifies the relationship between dwarfing and N utilization. Therefore, future studies are needed to explore the other mechanism of DELLA-N in pear (Line 343-348).
